# Silicon Carbide-Gated Nanofluidic Membrane for Active Control of Electrokinetic Ionic Transport

**DOI:** 10.3390/membranes11070535

**Published:** 2021-07-15

**Authors:** Antonia Silvestri, Nicola Di Trani, Giancarlo Canavese, Paolo Motto Ros, Leonardo Iannucci, Sabrina Grassini, Yu Wang, Xuewu Liu, Danilo Demarchi, Alessandro Grattoni

**Affiliations:** 1Department of Electronics and Telecommunications, Polytechnic of Turin, 10129 Turin, Italy; antonia.silvestri@polito.it (A.S.); paolo.mottoros@polito.it (P.M.R.); danilo.demarchi@polito.it (D.D.); 2Department of Nanomedicine, Houston Methodist Research Institute, Houston, TX 77030, USA; nditrani@houstonmethodist.org (N.D.T.); ywang2@houstonmethodist.org (Y.W.); xliu@houstonmethodist.org (X.L.); 3Department of Applied Science and Technology, Polytechnic of Turin, 10129 Turin, Italy; giancarlo.canavese@polito.it (G.C.); leonardo.iannucci@polito.it (L.I.); sabrina.grassini@polito.it (S.G.); 4Department of Surgery, Houston Methodist Research Institute, Houston, TX 77030, USA; 5Department of Radiation Oncology, Houston Methodist Research Institute, Houston, TX 77030, USA

**Keywords:** ionic modulation control, nanofluidic ion transport, ion rectification, nanofluidic channel, electrical double layer

## Abstract

Manipulation of ions and molecules by external control at the nanoscale is highly relevant to biomedical applications. We report a biocompatible electrode-embedded nanofluidic channel membrane designed for electrofluidic applications such as ionic field-effect transistors for implantable drug-delivery systems. Our nanofluidic membrane includes a polysilicon electrode electrically isolated by amorphous silicon carbide (a-SiC). The nanochannel gating performance was experimentally investigated based on the current-voltage (I-V) characteristics, leakage current, and power consumption in potassium chloride (KCl) electrolyte. We observed significant modulation of ionic diffusive transport of both positively and negatively charged ions under physical confinement of nanochannels, with low power consumption. To study the physical mechanism associated with the gating performance, we performed electrochemical impedance spectroscopy. The results showed that the flat band voltage and density of states were significantly low. In light of its remarkable performance in terms of ionic modulation and low power consumption, this new biocompatible nanofluidic membrane could lead to a new class of silicon implantable nanofluidic systems for tunable drug delivery and personalized medicine.

## 1. Introduction

Advances in biomedical engineering aiming at new solutions for personalized medicine have fostered new developments in fields ranging from interface electronics to biological systems [1]. In this context, nanofluidic devices have been increasingly used for control and manipulation of ionic species [2], energy conversion [3,4,5,6], ionic devices [7,8,9], biosensing [10], analysis and separation of biomolecules [11,12], and drug delivery [1,13,14]. Along with this, solid-state technological processes for very large-scale integration (VLSI) devices have enabled the fabrication of devices with high resolution and high-aspect-ratio features such as nanometric channel membranes [9,13,14,15,16,17]. Among these systems, nanofluidic structures such as nanopores [18,19,20], nanoslits [21,22,23], and nanochannels [24,25,26] exhibit electrokinetic phenomena, such as ion current rectification, ion depletion, and ion gating, through the manipulation of their permselectivity. Due to their molecular feature size, nanofluidic channels can offer precise control of and interaction with charged species, opening the possibility to develop new biomedical devices and therapeutic platforms for personalized medicine as well as chronotherapeutic regimens for chronic and dysfunctional diseases [27,28,29]. The active interaction of charged surfaces with aqueous media containing charged species such as ions [30,31], DNA [32,33], proteins [7], and nanoparticles [34] permits the implementation of analysis and processing for biosensors, molecular separation techniques, and drug-delivery systems [13,14,27,28,29]. This electrostatic interaction results in accumulation of ionic species at the surface of the nanofluidic channels, known as the electrical double layer (EDL). These ions, called counterions, exhibit an opposite charge with respect to the nanofluidic surface charge, and their organization forms an electrostatic shield. This phenomenon results in electrostatic repulsion of the ions exhibiting the opposite charge of the counterions, called coions. The presence of the EDL causes the electric potential to decay to its bulk value over the characteristic length known as the Debye length λ (~1–100 nm). Then, modifying the surface charge through an electrostatic external stimulus such as an electric field leads to alteration of the EDL thickness. This phenomenon is widely used in nanofluidic structures to control the motion of the charged species. At present, several devices for electrostatic control of ionic species have been developed, such as ionic transistors [7,8,17], nanofluidic membranes with applied external electrodes [16,35], and anodic aluminum oxide (AAO) membranes [36]. These applications successfully demonstrated variations in the ion permselectivity of the nanochannels through exclusion enrichment [37] and concentration polarization [38] by applying an electrostatic field [39]. However, the limited ionic flow offered by ionic transistors is not suitable for drug-delivery applications; their channel density should be increased to reach an appropriate release rate [17]. Additionally, AAO membranes present a gate control localized at one end of the nanofluidic channels and a dispersed pore size distribution of the nanochannel due to the fabrication process, which can negatively affect fine control of release rate modulation. Our previous devices successfully modulated the extension of the EDL through electrostatic gate control applied with an embedded electrode buried on one side of the nanochannel [17] or external electrodes positioned on both sides of the membrane [35]. Nevertheless, the one-side electrode structure could alter the EDL only on one side of the nanofluidic structure. For this reason, the extension of the EDL over the entire nanochannel required a high electric field. In contrast, in the case of external electrodes, the nonintegrated metallic structure could be subjected to mechanical issues, such as detachment from the application site, leading to a lack of gate control.

The biomedical application for implantable drug delivery poses another challenge. The materials used to fabricate nanofluidic devices are crucial to the biocompatibility of the devices. Several studies were performed on different materials that could be used in biomedical applications [40,41,42,43]. a-SiC demonstrated higher mechanical stability, chemical inertness, and no evidence of corrosion under physiological conditions compared to SiO_2_ [15] and silicon nitride (Si_3_N_4_) [44]. Consequently, it is a suitable material for long-term implantable applications.

This study investigates the performance of a new gated nanofluidic membrane for electrostatic control of charged species for implantable applications of drug-based devices. Based on our knowledge, we created a nanofluidic membrane with a doped polysilicon buried gate electrode coated with amorphous silicon carbide (a-SiC) [15,45]. Using photolithographic processes widely employed in bio-nanoelectromechanical (BioNEMS) [46] and metal-oxide-semiconductor field-effect transistor (MOSFET) technologies, we achieved a high-density nanochannel-packed membrane that features a gate electrode deposited all around the nanochannel surface. The gating performance was evaluated by applying an electrostatic potential (−3 V to 3 V) to the gated nanochannels. The I-V characteristics were measured in 10^−5^ M potassium chloride (KCl) aqueous solution to assess the ionic FET (IFET) permselectivity of the nanofluidic membrane. Then, we evaluated the performance at higher concentrations (10^−4^ M KCl), where the ionic Debye length is smaller than the nanochannel size, in light of future biomedical applications such as reservoir-based implants. Furthermore, the study enabled identification of an asymmetric leakage current while performing electrostatic modulation of the EDL as well as estimation of the power consumption. To further investigate the physics of the devices, we inspected the behavior of the gated electrode of the nanochannel by reproducing it on a blank device, and we performed electrochemical impedance spectroscopy (EIS). We fitted the EIS measurements using a small-signal linear time-invariant (LTI) circuit, and we identified the transport phenomena responsible for the leakage current. Additionally, we evaluated the flat band potential and the density of the interface states. These findings can help in the design of nanofluidic membranes for fine-tunable electrostatic control of the drug release rate for implantable drug-based devices. The outcome of this investigation may lead to a new generation of silicon nanofluidic devices for implantable biomedical applications using VLSI technology.

## 2. Materials and Methods

### 2.1. Nanofluidic Membrane Fabrication

As previously described elsewhere [15], the lithographic fabrication process was performed on a silicon-on-insulator (SOI) 4-inch (100)-oriented p-doped wafer with a device layer (15 ± 1 µm), a buried oxide layer (1 µm), and a handle wafer (400 µm, Ultrasil Corporation, Hayward, CA, USA). After a first wet thermal oxidation step to produce 600 nm of sacrificial oxide on the surface of the device layer, nanoslit templates with a 500-nm width and a 6-µm length were patterned by using a reactive ion etching (RIE) step via inductively coupled plasma (ICP) deep silicon etching. On the other side of the SOI wafer, the handle wafer was patterned with a high density of hexagonally arranged circular microchannels by ICP to provide mechanical stability. Then, the built-up polymer covering the nanochannels and microchannels was cleaned, and the buried thermal oxide layer was removed in a buffered oxide etchant (BOE) solution. Nanochannels were shrunk by growing a layer of SiO_2_ through wet thermal oxidation. To build the buried gate electrode, a low-pressure chemical vapor deposition (LPCVD) step allowed the deposition of 121 nm of phosphorus-doped polysilicon (poly-Si) over the surface of the nanochannels. A further plasma-enhanced chemical vapor deposition (PECVD) step coated all of the wafer with a 64-nm a-SiC dielectric layer. Two contact pads (1 × 1 mm^2^) were created by selective removal of a-SiC by fluorine-based RIE at the edge of the membrane to expose the doped poly-Si. One hundred twenty total membranes were diced into 6 × 6 mm^2^ pieces using a dicing saw. Each membrane presents 199 microchannels organized in a hexagonal spatial configuration. Each microchannel is connected to 1400 identical slit nanochannels arranged in 19 rows and 96 columns, and consequently, the membrane chip features 278,600 nanochannels.

### 2.2. Blank Fabrication

A planar gate electrode was reproduced on the same type of wafer by following the same photolithographic process. Briefly, a 4-inch p-doped SOI wafer underwent a wet thermal oxidation step, producing a layer of SiO_2_. Then, the gate electrode was obtained via LPCVD of 121 nm of phosphorus-doped poly-Si. The wafer was diced in half before proceeding to coat it with the dielectric. On half of the wafer, a PECVD step deposited an a-SiC dielectric coating, and on the other half of the wafer, a wet thermal oxidation step produced a SiO_2_ coating. The contact pads for the poly-Si were created by selective removal of the dielectric by fluorine-based RIE at the edge of the blanks. Each half wafer hosted sixty 6 × 6 mm^2^ blanks diced via a dicing saw.

### 2.3. Electrode Connection

One contact pad exposing the poly-Si was connected to an insulated high-temperature 36 AWG wire via conductive silver epoxy (H20E, Epoxy Technology, Billerica, MA, USA) and underwent a 1-h curing step at 150 °C. The conductive exposed connection was isolated with UV epoxy (OG116, Epoxy Technologies, Inc., Billerica, MA, USA) and cured via UV for 2 h.

### 2.4. Setup for Electrochemical Measurements

An illustration of the setup for electrochemical measurements is reported elsewhere [15]. Electrochemical measurements were performed in a poly (methyl methacrylate) (PMMA, McMaster Carr, Douglasville, GA, USA) fixture. The setup system was constituted by two reservoirs containing 2 mL of solution, each with two Ag/AgCl electrodes (counter electrode and reference electrode). The samples under investigation (membranes and blank) were secured between the two reservoirs by rubber silicone O-rings (Apple Rubber, Lancaster, NY, USA). The fixture was screwed via 4 SS316L M3 screws. All measurements were performed using a multichannel potentiostat (Ivium Technologies B.V., Eindhoven, The Netherlands).

### 2.5. Scanning Electron Microscopy (SEM), Atomic Force Microscopy (AFM), and Ellipsometry

To assess the nanofluidic membrane structure, we morphologically inspected nanochannels by obtaining cross-sections of the nanofluidic membrane via a gallium ion milling-focused ion beam (FIB, FEI Dual-Beam 235 FIB, FEI, Hillsboro, OR, USA). The size of the nanochannels was measured with SEM (Nova NanoSEM 230, FEI, Hillsboro, OR, USA).

To investigate the thickness of the deposited layer, we performed ellipsometry measurements using a J.A. Woollam M2000U ellipsometer (Lincoln, NE, USA) on a-SiC and SiO2 blanks. To assess the roughness of the surface, we performed topographic mapping by AFM (Bruker MultiMode Atomic Force Microscope, Billerica, MA, USA).

### 2.6. Conductance, Current-Voltage (I-V) Curves, and Leakage Current

Electrochemical characterization of the nanochannels was performed in a KCl solution. The conductance was measured in a 4-electrode configuration using different concentrations of KCl, ranging from 10^−7^ M to 10^1^ M. A potential across the membrane, also called the transmembrane potential (V_DS_), was applied using one channel of the potentiostat (Ivium Multichannels). To monitor the transmembrane current I_DS_, a staircase of 250 mV steps was applied from −1 V to 1 V in a 4-electrode configuration, and each step was held for 30 s to overcome transient phenomena. The conductance was calculated by taking the last sample point of each step and averaging the measured values. We repeated the measurements per membrane three times, and the complete study was performed on three membranes. During the conductance measurements, a gate potential was not applied.

The I-V characteristics and leakage current were evaluated in 10^−5^ M KCl and 10^−4^ M KCl. A voltage across the membrane (V_DS_) and a voltage on the gate electrode of the membrane (V_GS_) were applied using two channels of the potentiostat (Ivium Multichannels) to monitor the transmembrane current I_DS_ and the leakage current I_GS_ in parallel. To monitor I_DS_, a staircase of 250 mV steps was applied from −1 V to 1 V in a 4-electrode configuration, and each step was held for 30 s to overcome transient phenomena. V_GS_ was applied between the gate electrode and two Ag/AgCl electrodes in the source reservoir facing the nanochannel layer device in a 3-electrode configuration with a DC potential ranging from −3 V to 3 V at a 500 mV step. I_DS_ and I_GS_ were measured under different V_DS_ and V_GS_ nested voltage sweeps applied to the working and reference electrodes.

### 2.7. EIS

EIS measurements were performed using blank samples with two different dielectrics, a-SiC and SiO_2_. The blank was inserted in the previous fixture, for which only the reservoir with two Ag/AgCl electrodes was used. The pressing of the blank samples against the cell via the O-ring left 7.07 mm^2^ exposed to the solutions. The measurements were performed in a 3-electrode configuration in 10^−4^ M KCl. Impedance spectra were obtained using a 50 mV V_PP_ amplitude AC signal over a frequency range from 10^−2^ Hz to 10^5^ Hz and a DC potential scan ranging from −3 V to 3 V, with a step of 500 mV. Mott-Schottky analysis was carried out at a 10^5^ Hz frequency to calculate the carrier concentrations, flat band potential, and density of interface states.

## 3. Results and Discussion

### 3.1. Nanofluidic Membrane

The final nanofluidic membrane was inspected via optical microscopy to assess the mechanical integrity and the average size of the nanochannels measured via nitrogen flow test [47] was estimated ~106 nm. The etching procedure and layer deposition uniformity were analyzed using FIB-SEM microscopy. Figure 1A displays the 6 mm × 6 mm × 0.4 mm nanofluidic membrane, highlighting the electrode pads at the top left and bottom right edges. The membrane features 199 vertically oriented microchannels arranged in a hexagonal fashion to optimize the membrane porosity, simultaneously preserving the structural stability. Figure 1B presents an SEM image of the nanochannels densely packed in 19 rows and 96 columns for each microchannel. The nanochannel cross-section is shown in Figure 1C, and a detailed nanochannel cross-section with different layers is presented in Figure 1D.

The image shows that the layers were deposited, achieving good thickness uniformity despite the high aspect ratio of the nanochannels. In comparison with our previously developed nanofluidic technologies [17,48,49,50], the nanofluidic membrane requires a simplified photolithographic process owing to the easy connection derived from the vertical orientation of both microchannels and nanochannels [16,50] and features a substantially higher nanochannel number [17,51].

Compared to other nanofluidic devices for gate control, such as AAO [36] membranes fabricated with the gate electrode only at one end of the nanochannels, our nanochannel membrane features an extended buried polysilicon gate electrode. The buried electrode represents a crucial aspect that allows fine electrostatic control by modulating the ionic conductance along the entire length of the nanochannels. Additionally, the streamlined structure that connects nanochannels and microchannels permits the development of a nanofluidic membrane that features 278,600 nanochannels, permitting a sustainable release rate in view of future biomedical applications for therapeutic administration [15].

### 3.2. Blanks

#### 3.2.1. Ellipsometry and AFM

The thickness of the deposited layers of the blank samples was assessed via ellipsometric measurements. Measurements related to a-SiC are presented in Figure 2A. The polysilicon layer in both cases was measured to be ~121 nm ± 15 nm, the a-SiC layer was measured to be ~64 nm ± 10 nm, and the SiO_2_ was measured to be ~66 nm ± 13 nm. Morphological evaluation of the surface of the dielectric coating was performed via AFM investigation. Measurements were performed at three different spots and averaged. Figure 2B shows a 50 μm × 50 μm AFM topographic map for an a-SiC blank device. The surface roughness was 22.9 nm ± 4.7 nm for a-SiC and 14.5 nm ± 3.9 nm for SiO_2_. The higher surface roughness measured for a-SiC blank devices can be associated with the pump-down vacuum phase verifiable during the fabrication process. During PECVD, the debris lying on the surface of the chamber could be redistributed on the sample surface by turbulent flows generated during the process.

### 3.3. EDL Modulation: Theoretical Background and Operation Mode

At the nanoscale, many transport processes in liquid are dictated by the interaction of ions with charged surfaces. Material surfaces exhibit an electrical charge when in contact with an aqueous medium where charged species are dissolved. To neutralize the electrical charges present on the surface, counterions electrostatically accumulate at the material/solution interface. Their distribution forms a layer called the EDL. Considering the Gouy-Chapman-Stern (GCS) model, the EDL is formed by two layers of ions. The first layer, called the compact layer, is composed of solvent molecules and ions that are immobile and adsorbed on the material–solution interface. The second layer positioned over the first layer is called the diffuse layer and consists of a mobile layer carrying a net charge within the solution. The thickness of the EDL depends on the ionic strength of the aqueous solution and the surface charge. In a nanochannel where the EDL thickness can be comparable to at least one of its sizes, the exclusion enrichment effect [37,52,53] and concentration polarization [38] can be observed. These phenomena can actively alter the diffusional flow of charged species and modify the ionic conductance in the nanochannel. Thus, modification of the surface charges leads to a variation in the EDL thickness and, consequently, ionic conductance. Control of the surface charge of a nanochannel can be obtained through application of an electrostatic field to the nanochannel wall, as shown in Figure 3. This phenomenon can be called electrostatic gate modulation [54]. Owing to this phenomenon, the surface charges of the nanochannels can be customized, allowing fine-tunable control of the ionic transport over the EDL thickness. Figure 3B shows the altered EDL thickness as a consequence of the electrostatic control applied to the nanochannel. To investigate the features of our nanofluidic membrane, we measured the conductivity to determine the surface charge of the nanochannels, while the gate control performance and permselectivity variation can be assessed through I-V measurements.

### 3.4. Ionic Conductance at the Floating Gate

To electrically characterize the nanochannels, we first performed nanochannel conductance measurements by applying transmembrane potential V_DS_ ranging from −1 V to 1 V with the gate electrode floating. Figure 4 shows the nanochannel conductance as a function of KCl concentration. The conductance characteristic presents two parts: the bulk conductance occurring when the nanochannel height (h) is larger than the Debye length (λ), with the ratio h/λ » 1; the surface-dominated conductance region occurring when h is smaller than λ, with h/λ ~ 1 or h/λ < 1. In the bulk conductance region, the conductivity is linearly dependent on the electrolyte concentration. In contrast, in the surface-dominated conductance region, the conductivity shows a plateau due to the enrichment of the counterions to compensate and neutralize the surface charge. For this reason, the conductance only depends on the surface charge.

The transition between the two regions occurs at 10^−5^ M, where the Debye length is ~96 nm. Considering the conductance of the nanochannels as the superimposition of the two contributions, the bulk conductance and the surface charge conductance, we can apply the following formula [21]:(1)$$G=10^{3}\left(\mu_{K^{+}}+\mu_{Cl^{-}}\right)cN_{A}e\frac{wh}{l}+2\mu_{K^{+}}\sigma \frac{w}{l}$$
where $\mu_{K^{+}}$ and $\mu_{Cl^{-}}\;$ are the ionic mobilities of K and Cl, c is the molar concentration, $N_{A}$ is Avogadro’s number, and *w*, *h*, and *l* are the width, height, and length of the nanochannel, respectively. Using the width and length obtained from the morphological characterizations, we evaluated the height of the nanochannels. By utilizing the bulk conductance formula, we obtained an average nanochannel height of ~106 nm. The results are consistent with the obtained gas flow test measurements [47]. Using the surface-dominated conductance formula (1), we calculated the surface charge on the nanochannel walls to be 0.471 μCm^−2^ at pH 7.4. This result is consistent with other surface charge values for SiC [55] and our previous studies [15,45].

The native surface charge exhibited by the a-SiC is smaller than that exhibited by the widely used SiO_2_ at pH 7.4 (1–100 mCm^−2^) [7]. This aspect provides benefits when gate control is needed. The reason lies in the need to electrostatically control the reaction at the interface that acts as a charge buffer. The ability to regulate the protonation and deprotonation of reactive groups present at the dielectric/electrolyte interface is key to generating and controlling the charge distribution in the double layer [56].

### 3.5. I-V Characteristics and Leakage Current

The nanofluidic membrane device has two distinguishing features: (1) phosphorus-doped polysilicon with an excess of electron carriers as the gate electrode and (2) low surface charge density. The first feature allows control of the leakage currents and consequently the power dissipation of the devices. Regarding the second aspect, exhibiting a low surface charge density environment can help with high-efficiency gate modulation. In fact, small changes in gate potentials can be relatively significant in the case of very low surface charge [57,58].

To test the performance of the nanofluidic membrane, ionic current I_DS_ vs. V_DS_ curves were measured as a function of V_GS_ within the voltage range of |V_GS_| < 3 V at a step of 0.5 V and |V_DS_| < 1 V at a step of 0.25 V. To evaluate the power consumption, we simultaneously measured the leakage current I_GS_ vs. V_GS_ flowing from the gate to source. To test the ability of the nanofluidic membrane to modify the permselectivity of the nanochannels, we performed measurements in 10^−5^ M KCl, the concentration at which the nanochannels present the transition between the surface-charge-governed regime and the geometry-governed regime. Here, the electrolyte exhibited a Debye length comparable to the nanochannel height of the nanofluidic membrane. Moreover, we evaluated the gating performance of the nanofluidic membrane using 10^−4^ M KCl, a concentration at which λ < h. The purpose of this investigation is to evaluate the capability of the n-doped polysilicon electrode to manipulate the extension of the EDL in the nanochannel under the condition of the conductance being dictated by the geometry.

Figure 5 shows a representation of the ionic flow when gate and transmembrane potentials are applied. The transmembrane potential V_DS_ influences the direction of the ions in the nanochannels. The application of V_DS_ < 0 V directs potassium ions K^+^ to the drain electrode and negative chloride ions Cl^−^ to the source electrode, whereas V_DS_ > 0 V conveys K^+^ to the source and Cl^−^ to the drain. Some of the ions in the nanochannels are partially involved in the formation of the EDL depending on the polarization potential applied to the gate electrode and consequently the exposed surface charge. The negative charge of the nanochannel wall and the n-doping profile of the polysilicon strongly influence the ionic transport in the nanochannel since it exhibits an excess of negative carriers (i.e., electrons in the polysilicon) that attract a majority of positive ions to the nanochannel wall. As a consequence, the ionic transport in the nanochannels and thus the conductance and leakage currents are strongly influenced.

Upon application of polarization potential V_GS_, the charged ionic species are electrostatically attracted to the nanochannel wall and participate in forming the EDL. V_GS_ < 0 V attracts positive K^+^ to the nanochannel wall, creating a positively charged EDL. As a consequence, the number of positive ions K^+^ in the nanochannel increases (Figure 5A,B), and negative ions Cl^−^ are repelled. In contrast, V_GS_ > 0 V attracts Cl^−^ ions to the nanochannel wall, creating a negative EDL and repelling K^+^ (Figure 5C,D).

The experimentally measured I_DS_-V_DS_ characteristics and leakage current I_GS_-V_GS_ in 10^−5^ M KCl and 10^−4^ M KCl are presented in Figure 6. The nanochannel membrane shows a unipolar behavior with a shallow surface charge density due to the a-SiC layer and the gate electrode, the so-called all-around-gate structure. Figure 6A,C shows I_DS_ for 10^−5^ M KCl and 10^−4^ M KCl, respectively. At higher KCl concentrations, we obtained higher ionic current values. When V_GS_ = 0 V, the ionic current shows a linear ohmic relationship between I_DS_ and V_DS_. The same ionic current is expected for both coions and counterions, independent of the drain and source electrode, because potassium and chloride have similar mobilities in a liquid (76.2 × 10^−7^ m^2^/sV for K^+^ and 79.1 × 10^−7^ m^2^/sV for Cl^−^) [59]. Upon applying the gate voltage, the ionic conductance can vary based on the polarity of the ions in the bulk.

In solid-state electronics such as FET devices, the electronic current depends on the gate voltage and the doping of the device, and the nanofluidic membrane exhibits behavior comparable to that of a p-type junction FET (JFET), where upon application of a positive gate voltage, the electronic current decreases. Application of V_GS_ > 0 V leads to reduction in the ionic conductance and I_DS_ over the range |V_DS_| < 1 V. In contrast, application of V_GS_ < 0 V leads to two different scenarios based on the polarity of V_DS_. For both concentrations 10^−5^ M and 10^−4^ M KCl, a negative transmembrane potential V_DS_ < 0 V causes increases in the ionic current and conductance in the nanochannels, whereas a positive transmembrane potential V_DS_ > 0 V generates decreases in the ionic current and conductance in the case of 10^−5^ M KCl and a slight effect on the transmembrane current I_DS_ and conductance in the case of 10^−4^ M KCl.

To explain the ionic transmembrane current behavior, we can refer to conduction mechanisms in an aqueous solution. The recombination process between the high number of electrons in the polysilicon and the ions in the aqueous solution [60,61] could be responsible for the different responses of the nanochannels. Additionally, we need to consider that a-SiC can act as an amorphous n-doped semiconductor [62]. Regarding this last aspect, the polarization potential V_GS_ can strongly influence the conduction mechanism in the dielectric a-SiC. Due to its nature, a-SiC favors recombination of electrons with positive ionic species. Realistically, in the case of V_GS_ < 0 V, some of the positive charges electrostatically attracted to the nanochannel wall could be involved in the conduction mechanism, and consequently, the thickness of the EDL could be reduced since the species return to their electroneutral state [60,61]. Upon application of V_DS_ < 0 V, many positive ions are attracted into the nanochannels, increasing I_DS_ and the conductivity. In contrast, V_DS_ > 0 V drives negative charges into the nanochannels that are repelled by the application of a negative polarization potential. In the case of V_GS_ > 0 V, the negative ions Cl^−^ attracted to the nanochannel wall are negligibly involved in the conduction mechanism. Higher values of V_GS_ induce the EDL to extend in the nanochannels, leading to reduced I_DS_ and conductance. Considering the I_DS_ in 10^−4^ M KCl for V_DS_ = 1 V, V_GS_ = 3 V causes I_DS_ to decrease from 600 nA to 129 nA, corresponding to a promising reduction of 78.5% in the ionic transmembrane flow. We also noted that for values of V_GS_ in the range of 1.5 V to 3 V, I_DS_ exhibits a slight reduction. This phenomenon can be associated with the electron depletion region formed at the polysilicon-dielectric interface. In fact, for a positive polarization potential, the electrostatic attraction of negative ions to the dielectric interface is achieved by the lack of electrons at the polysilicon-dielectric interface. This lack produces an electron depletion region, where positive electronic carriers (called holes) are responsible for electrostatic control of the EDL. Since the polysilicon is doped with an excess of electrons, holes are numerically inferior in the depletion region. Therefore, the positive electrostatic charge that can be modulated in the depletion region is lower than the negative charge originating from negative V_GS_. As a consequence, the EDL cannot be exceedingly extended.

The n-doping of the polysilicon and the behavior as a negative amorphous semiconductor of the a-SiC are responsible for the asymmetric leakage currents presented in Figure 6B,D. In fact, these two aspects strongly influence the conduction mechanism upon voltage application, working in favor of the conduction mechanism for negative gate potential V_GS_ and against it for positive gate potential V_GS_. Further details are presented in the next section.

Figure 7 shows the nanochannel conductance for 10^−4^ M KCl as the ratio of the conductance modulated by the application of V_GS_ and the conductance at the floating gate at V_DS_ = −1 V and V_DS_ = 1 V. The graph indicates that for V_DS_ = −1 V, the conductance is enhanced for V_GS_ < 0 V and reduced for V_GS_ > 0 V. In the case of V_DS_ = 1 V, V_GS_ < 0 V does not significantly influence the conductance in the nanochannels, while V_GS_ > 0 V causes a significant reduction in the normalized conductance. Additionally, the conductance exhibited at V_GS_ = 3 V is barely reduced compared to the conductance calculated at V_GS_ = 1.5 V. This finding is in agreement with the previously described ionic transport phenomena occurring in the nanochannels.

Leakage currents allow the estimation of the power consumption (P_diss_) of the nanofluidic membrane. The worst working conditions are created when I_GS_ exhibits higher values in the range of applied gate potential of −3 V to 3 V. Referring to 10^−4^ M KCl, higher values of I_GS_ are shown for V_GS_ = ± 3 V and I_DS_ = 0 V. In these cases, the power consumption (P_diss_) results in P_diss_ = 4.2 μW for V_GS_ = −3 V and P_diss_ = 1.74 μW for V_GS_ = 3 V. By applying V_DS_, Pdiss for V_GS_ = 3 V can be reduced by approximately 22%, and in the case of V_GS_ = −3 V, the power consumption can be reduced by 87%. Compared with previous work, the power consumption was reduced by one order of magnitude [15]. This can be related to the fact that the ionic strength of the solution used for this measurement is 1.37 times lower than the ionic strength of the solution used for our previous study [15].

An advantageous aspect of having asymmetric leakage currents lies in reducing the power consumption. In fact, we meet the need to reduce the ionic conductance by modulating the EDL extension in the nanochannel with the application of V_GS_ > 0 V. Under this condition, the considerable performance of the nanofluidic membrane is associated with low power consumption.

The investigation of the capability of the nanofluidic membrane to electrostatically modulate the conductance and ionic transport in the nanochannel geometry-dictated regime can lay the groundwork to use biocompatible nanofluidic membranes in reservoir-based implants. In such devices, the highly concentrated drug in the reservoir is released at a rate established by the geometrical dimensions of the nanochannels [27,28,29,50]. In view of future manipulation of charged drug molecules, evaluation of the electrostatic control of ionic species exhibiting a Debye length λ almost one-third of the nanochannel height can provide a demonstration of the performance of the nanofluidic membrane.

### 3.6. EIS

To further investigate the physics of the nanofluidic membrane, we focused on the gate electrode interface immersed in 10^−4^ M KCl aqueous solution. To do so, we reproduced the dielectric-polysilicon heterojunction on a blank device, and we performed EIS. We conducted the same investigation on blanks with SiO_2_, comparing the dielectric performances. Measurements were performed in a frequency range from 10^−2^ Hz to 10^5^ Hz by applying D.C. polarization potentials to the gate electrode from −3 V to 3 V at a step of 0.5 V. These bias conditions hold the structure at an operating bias point such that the behavior of the structure is fairly linear over a small range of voltages around the bias point. First, we determined the transport properties associated with the heterojunction immersed in KCl and associated a small-signal LTI electric circuit and the transfer function. The transfer function represents an essential aspect of designing electronic control customized based on physical phenomena between solids and electrolytes. Then, we calculated the flat band potential V_fb_ by using the Mott-Schottky plot. Compared with the SiO_2_, the a-SiC exhibits a lower V_fb_, which is favorable for easily manipulating the surface charge. Then, we calculated the density of states *D_it_*. These are “defects” that can be responsible for a higher leakage current and can be associated with the chemical process used to fabricate the dielectric/polysilicon interface.

#### 3.6.1. Band Diagram Theory

When immersed in an aqueous solution, the blanks and the solution form a system that is organized into three layers and two interfaces: the first layer corresponds to the semiconductor n-polysilicon, which represents the buried electrode, the second layer corresponds to the dielectric coating, a-SiC or SiO_2_, and the third layer corresponds to the aqueous solution, as shown in Figure 8A. This is very similar to the solid-state MOS capacitor device composed of a metal layer, a dielectric layer, and a polysilicon layer. Here, the metal layer is replaced by the conductive electrolyte solution KCl, and the gate voltage is applied to the polysilicon. To explain how the polarization potential influences the carriers in the polysilicon, we can refer to the band theory. Application of a polarization potential puts the blank into one of three states: accumulation, flat band condition, and inversion, as illustrated in the band diagram presented in Figure 8B. When no polarization potential is applied with V_GS_ = 0 V, the Fermi level E_F_ in the polysilicon is pinned close to the conduction band E_C_ due to the nature of the phosphorus donors. Notably, there is no Fermi level in the solution in contact with the dielectric since there is no redox couple in it. Applying a negative polarization potential V_GS_ < 0 V to the electrode causes electron accumulation at the dielectric-semiconductor interface. The band bending presents a downward curvature and a thin layer, typically a few angstroms, of high electron concentration. This phenomenon leads to positive ions accumulating at the dielectric-electrolyte interface. Applying a positive polarization bias (V_GS_ > 0) to the gate electrode causes a region depleted of electrons called the depletion zone to be formed. Here, the majority of the carriers are holes, which are the native charges of the semiconductor, and the band bending presents an upward curvature. Mainly, in both cases, band bending occurs in the semiconductor, and a small part of it occurs at the double layer. Although the chemical potential in the dielectric and electrolyte cannot be well defined, the electrostatic potential drop can be determined [63]. For a capacitor, V = q/C, the potential drop across the EDL is smaller than the potential drop across the dielectric (C_EDL_ > C_SD_). According to the theory of solid-state semiconductor devices such as MOS capacitors [64], in the accumulation region, the capacitance increases due to the high number of electrons. Meanwhile, in the depletion region, the lack of electrons causes the capacitance to decrease until its value becomes almost constant.

#### 3.6.2. Schematic Circuit

The impedance spectra of the data as Nyquist plots measured on a-SiC blank device are reported in Appendix A. An equivalent LTI model for the buried electrodes was proposed and is shown in Figure 8C. The equivalent circuit includes a combination of resistances (R) and capacitors (C). For interpretation of the results, the fact that the resistance is a parameter strongly related to the carrier transport properties must be considered, and at the same time, the capacitance is linked to the carriers. The circuit is composed of the following elements: R_M_ represents the resistance of the electrochemical solution 10^−4^ M KCl. The first parallel part is associated with the electrolyte-dielectric interface, and it is composed of C_EDL_, the capacitance associated with the EDL, and R_CT_, the charge transfer resistance associated with the charge transfer through the dielectric coating. The second parallel part is associated with the dielectric-semiconductor interface, and it is composed of C_SD_, the capacitance associated with the semiconductor interface state capacitance, and R_SD_, the resistance associated with the semiconductor interface state resistance.

We considered the nonideality of the interfaces, taking into account the nonconstant thickness of the deposited layers (polysilicon layer ~121 nm ± 15 nm, a-SiC layer ~64 nm ± 10 nm, and SiO_2_ layer ~66 nm ± 13 nm) and the experimental roughness (a-SiC layer ~22.9 nm ± 4.7 nm and SiO2 layer ~14.5 nm ± 3.9 nm) measured as reported in Section 3.2.1 due to the deposition process and chemical defects that can be present in both the semiconductor polysilicon and dielectric layers [65,66,67,68]. For this reason, we used a constant phase element to relate to the *i*-th capacitance via the relationship $Z_{CPEi}=\frac{1}{Q_{i}{\left(j\omega \right)}^{ni}}$, where *Q_i_* is the constant associated with the constant phase element, and ni is the empirical exponent measuring the distortion from the ideal impedance components. The constant phase element can be considered a capacitance when the value of *n_i_* satisfies 0.5 < *n* < 1. Figure 9 shows the R_CT_, R_SD_, Q_EDL_, and Q_SD_ for both 64 nm a-SiC and 64 nm SiO_2_. The resistance associated with the electrolyte solution R_M_ was estimated from the Nyquist plot and was found to be ~100 Ω in all measurements. Values were obtained by fitting the measured data with the circuit model shown in Figure 8C. Compared to the case without application of a polarization potential, both dielectrics SiC and SiO_2_ exhibit the same trend. Negative values of polarization potential V_GS_ < 0 V cause R_CT_ and R_SD_ to decrease, while Q_EDL_ and Q_SD_ increase. For positive polarization potential V_GS_ > 0 V, the opposite trend for R_CT_, R_SD_, Q_EDL_, and Q_SD_ can be observed. We associated this behavior with the accumulation or depletion of electrons at the dielectric-polysilicon interface, as previously illustrated. According to band diagram theory, as previously reported, in the accumulation state occurring for V_GS_ < 0 V, the high number of electrons at the polysilicon-dielectric interface leads to a higher availability of electrons that can participate in the transfer process from the polysilicon to the aqueous solution through the dielectric, leading to a decrease in resistances associated with the interfaces (Figure 9A,B). At the same time, this phenomenon increases the capacitance at the polysilicon-dielectric interface and consequently increases the capacitance of the EDL at the dielectric-electrolyte interface (Figure 9C,D). Analogously, in the depletion state, electron carriers are depleted. This phenomenon leads to increases in the interface resistances and decreases in the capacitances associated with both interfaces, polysilicon-dielectric and dielectric-electrolyte, with respect to the case of no polarization potential applied.

Compared with SiO_2_ at V_GS_ = 0 V, the a-SiC dielectric shows an R_CT_ hundreds of times smaller. By applying a polarization potential, considering the entire range of −3 V to 3 V, the R_CT_ for a-SiC maintains the trend, and it remains between tens and hundreds of times smaller than the RCT for SiO_2_. Analogously, the a-SiC R_DS_ is tens of times smaller than that of SiO_2_. Regarding the constant phase elements associated with the two electrolyte/dielectric and dielectric/semiconductor interfaces, n_EDL_ and n_SD_ are in the range of 0.7 to 1. This fact confirms that the constant phase element acts as a capacitance. The Q_EDL_ values exhibited by a-SiC are 1.5 to 8 times larger than those for SiO_2_. Very similar, the Q_SD_ values of a-SiC are 0.8 to 3.1 times larger than those of SiO_2_. We associated these differences with the fact that SiO_2_ is an insulator, while a-SiC can be considered as an amorphous semiconductor material. In amorphous materials, there are free immobilized electrons in the disordered network [69]. Their presence provides stored charges that increase the capacitance associated with these materials.

The corresponding relaxation times for the two interfaces can be determined as $\tau_{EDL={\left(Q_{EDL}R_{CT}\right)}^{n_{EDL}}}$ and $\tau_{DS={\left(Q_{DS}R_{DS}\right)}^{n_{DS}}}$. These quantities correspond to the amount of time needed for polysilicon and the dielectric to reach equilibrium. They are related to the charge transfer phenomena and to the recombination lifetime of the carriers. Specifically, for a-SiC, $\tau_{EDL}$ is between 3 ms and 1.8 s, while in the case of SiO_2_, $\tau_{EDL}$ is between 50 ms and 30 s, where the maximum values are measured at V_GS_ = 0 V. These values are in agreement with the theoretical explanation of amorphous semiconductor and insulator materials [70,71]. Regarding the recombination lifetime $\tau_{DS}$, a-SiC exhibits values between 8.9 μs and 12 μs, while SiO_2_ exhibits values between 38 μs and 43 μs. The measured values are in agreement with other values measured for polycrystalline silicon [72].

The total impedance exhibited by the buried electrode immersed in the ionic solution associated with the fitting electrical circuit (Figure 8C) can be written as:(2)$$Z=R_{M}+{\left(\frac{1}{R_{CT}}+\frac{1}{Z_{CPE}{}_{EDL}}\right)}^{-1}+{\left(\frac{1}{R_{DS}}+\frac{1}{Z_{CPE}{}_{DS}}\right)}^{-1}$$

The real and imaginary parts of the impedance can be separated and written as:(3)$$Re[Z]=R_{M}+\frac{R_{CT}(1+R_{CT}Q_{EDL}\omega^{n_{EDL}}\cos\frac{n_{EDL}\pi }{2})}{1+2R_{CT}Q_{EDL}\omega^{n_{EDL}}\cos\frac{n_{EDL}\pi }{2}+{\left(R_{CT}Q_{EDL}\omega^{n_{EDL}}\right)}^{2}}+\frac{R_{SD}(1+R_{SD}Q_{SD}\omega^{n_{SD}}\cos\frac{n_{SD}\pi }{2})}{1+2R_{SD}Q_{SD}\omega^{n_{SD}}\cos\frac{n_{SD}\pi }{2}+{\left(R_{SD}Q_{SD}\omega^{n_{SD}}\right)}^{2}}$$
(4)$$Im[Z]=\frac{R_{CT}^{2}Q_{EDL}\omega^{n_{EDL}}\sin\frac{n_{EDL}\pi }{2}}{1+2R_{CT}Q_{EDL}\omega^{n_{EDL}}\cos\frac{n_{EDL}\pi }{2}+{\left(R_{CT}Q_{EDL}\omega^{n_{EDL}}\right)}^{2}}+\frac{R_{SD}^{2}Q_{SD}\omega^{n_{SD}}\cos\frac{n_{SD}\pi }{2}}{1+2R_{SD}Q_{SD}\omega^{n_{SD}}\cos\frac{n_{SD}\pi }{2}+{\left(R_{SD}Q_{SD}\omega^{n_{SD}}\right)}^{2}}$$

#### 3.6.3. High Frequency

The flat band potential is determined from the polysilicon-dielectric interface, and it is associated with the change in the working state of the electrode. Electrical transport properties such as the current flow through the dielectric are associated with the charge “trap” sites at the polysilicon-dielectric interface. These sites are responsible for capturing electrons from polysilicon, generating an electron-depletion region in polysilicon and increasing the availability of electrons that can participate in the transfer processes with the electrolyte [64]. To understand the amount of charges available to participate in the conduction transfer process, we analyzed the resistance and capacitance at 10^5^ Hz. The chosen frequency corresponds to a short time scale that prevents filling and unfilling of the surface states and, consequently, building up of the double-layer capacitance. For this reason, the circuitry can be simplified as a single capacitor and a single resistor in parallel [63,73]. The capacitance-voltage (C-V) (Figure 10A) and conductance-voltage (G-V) (Figure 10B) values were used to determine the flat band potential, the number of donors, and the interface trap density.

The capacitance, C, and the conductance, G, were evaluated using the relationships
(5)$$G=\frac{Re\left[Z\right]}{Re{\left[Z\right]}^{2}+Im{\left[Z\right]}^{2}}$$
and
(6)$$C=\frac{-Im\left[Z\right]}{\omega \left(Re{\left[Z\right]}^{2}+Im{\left[Z\right]}^{2}\right)}$$
at 10^5^ Hz. *Im*[Z] and *Re*[Z] are the real and imaginary parts of the impedance Z, respectively. The conductance and capacitance results (Figure 10) confirm that in the accumulation state, the high number of carriers favor higher C and G values, while in the depletion state, the lack of carriers leads to lower C and G values.

To determine the flat band voltage V_fb_ and the number of donors N_D_, we used the Mott–Schottky plot presented in Figure 11A for the polarization potential bias window of −3 V to 3 V. According to Mott–Schottky theory, the capacitance and potential are related by the equation
(7)$$\frac{1}{C^{2}}=\frac{2}{\varepsilon \varepsilon_{0}A^{2}eN_{D}}\left(V-V_{fb}-\frac{k_{B}T}{e}\right)$$
where C is the interfacial capacitance, A is the area exposed to the electrolyte, N_D_ is the number of donors or carrier concentration, V is the applied voltage, V_fb_ is the flat band potential, k_B_ is the Boltzmann constant, T is the absolute temperature, and *e* is the electronic charge.

We evaluated the carrier concentration N_D_ and V_fb_ by taking the horizontal intercept of the 1/C^2^ versus V plot [64]. We estimated V_fb_ ~−0.23 V for 64 nm a-SiC and V_fb_ ~−0.70 V for 64 nm SiO_2_. We calculated the number of donors to be N_D_ ~2.8 × 10^15^ cm^−3^ for a-SiC and N_D_ ~5.67.8 × 10^14^ cm^−3^ for SiO_2_. The flat band voltage V_fb_ indicates the potential needed to neutralize the surface potential at the dielectric/electrolyte interface. The V_fb_ of the a-SiC blank is lower than that estimated for SiO_2_. As a consequence, the surface charge exhibited by the a-SiC is lower than that exhibited by the SiO_2_, confirming the previous results of a low surface charge exhibited by a-SiC in Section 3.4. This result represents a positive aspect since a positive surface potential can be switched to a negative surface potential by applying a low potential to the polysilicon. Regarding the number of donors, the slight difference found between a-SiC and SiO_2_ could be due to the roughness of the surface area (reported in Section 3.2.1) exposed to the electrolyte.

The density of interface states, shown in Figure 11B, was extracted for a-SiC and SiO_2_ from the G-V [74] data in the depletion [63] state using the equation [75]
(8)$$\frac{G}{\omega }=\frac{e\omega \tau }{1+{\left(\omega \tau \right)}^{2}},$$
where τ represents the recombination lifetime, evaluable as the product of the measured charge transfer resistance R_CT_ and the dielectric/polysilicon capacitance C_SD_ at the interface, $\tau =R_{CT}C_{SD}$.

The density of states indicates the general distribution of the electronic states in terms of energy. It is helpful for understating the conductivity and electrical transport phenomena at the interfaces. The D_it_ calculated for a-SiC and that for SiO_2_ exhibit similar trends (Figure 11B), and the values are low compared with other previously conducted studies [76]. The reason lies in the low ionic strength of the electrolyte used.

To evaluate the charge involved in the electrical transport mechanism, we can use the formula [77]
(9)$$Q_{it}=-e^{2}D_{it}V,$$
where *e* is the electron charge and V is the applied potential. The interface charge density values in the depletion state range from −1.6 × 10^9^ e cm^−2^ to −3.56 × 10^9^ e cm^−2^. SiO_2_ shows *Q_it_* values slightly higher in the range of −2.18 × 10^9^ e cm^−2^ to −3.28 × 10^9^ e cm^−2^. Due to the direct relationship with the density of states previously evaluated, the charges involved in the transport phenomena are fewer because of the corresponding low availability of charged ionic species in the electrolytic solution.

## 4. Conclusions and Future Outlook

In this paper, we have investigated the ability of a nanofluidic membrane to electrostatically control charged ions in a monovalent aqueous KCl solution through the application of a potential to the embedded electrode. The innovative approach implies a uniformly distributed doped polysilicon electrode buried in membrane nanochannels coated with biocompatible a-SiC. Using BioNEMS manufacturing techniques, an implantable nanofluidic membrane for fine-tunable electrostatic control was fabricated. The DC gating behavior of the nanochannel membrane was initially investigated through the I-V characteristics using 10^−5^ M KCl, which exhibits λ comparable to the nanochannel height. Furthermore, gating performances were investigated in the geometry-dictated regime using 10^−4^ M KCl. Depending on the polarity of the polarization potential V_GS_, the formed EDL favors or hinders ionic transport in the nanochannels. Considering a concentration of 10-4 M KCl, at V_DS_ = −1 V, the nanochannel conductance was enhanced by 1.5 times for V_GS_ = −3 V and reduced to 0.8 times for V_GS_ = 3 V with a low power consumption ranging from 1.74 μW to 4.2 μW. AC electrochemical properties were determined using EIS. By using band diagram theory and associating an LTI electrical circuit with the gated electrode, electrical transport phenomena at the dielectric-electrolyte interface were investigated.

This study demonstrated that the nanofluidic membrane can noticeably control ionic species exhibiting a λ that is one-third of the nanochannel height, thus in a geometry-dictated regime. The proven performances could be very useful in reservoir-based implantable devices, where the drug molecules could exhibit a λ smaller than the nanochannel dimensions due to the high concentration in the drug reservoirs. In this case, the molecules would be transported in the nanochannels in a geometry-dictated regime, as in our study. Additionally, the investigation of electrical transport phenomena at the dielectric-electrolyte interface demonstrated that phosphorus-doped polysilicon favors the leakage current for a negative polarization potential due to the accumulation of electrons at the polysilicon-dielectric interface. Further studies need to be conducted to investigate the behavior of the nanochannel membrane when using a p-doped semiconductor electrode. In this case, the acceptor mobility is typically lower in a p-type semiconductor than the donor mobility in the n-type semiconductor, which would reduce the leakage current.

## 5. Patents

Gated Nanofluidic Valve for Active and Passive Electrosteric Control of Molecular Transport and Methods of Fabrication, U.S. Provisional Pat. Ser. No. 62/961,437, filed 15 January 2020.

## Figures and Tables

**Figure 1 membranes-11-00535-f001:**
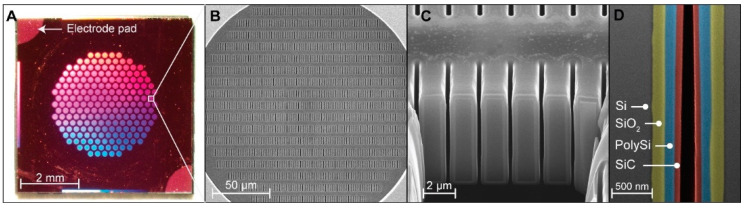
Image of the silicon nanofluidic membrane. (**A**). Final diced membrane with a size of 6 mm × 6 mm × 0.4 mm. In the photograph, the exposed contact pads of the polysilicon electrode used for connection to the external source for the polarization potential are indicated. (**B**) SEM image of the highly densely packed parallel nanochannel design. (**C**) FIB-SEM vertical cross-section showing the membrane nanochannels. (**D**) FIB-SEM vertical cross-section image of the wall of a nanochannel. The layers obtained at different stages of the photolithographic processes were color-enhanced to highlight the native original p-Si of the device layer of the wafer (in gray), thermally grown silicon dioxide (SiO_2_, ~175 nm, in green), poly-Si deposited by LPCVD (~121 nm, in blue), and a-SiC deposited by PECVD (64 nm, in red).

**Figure 2 membranes-11-00535-f002:**
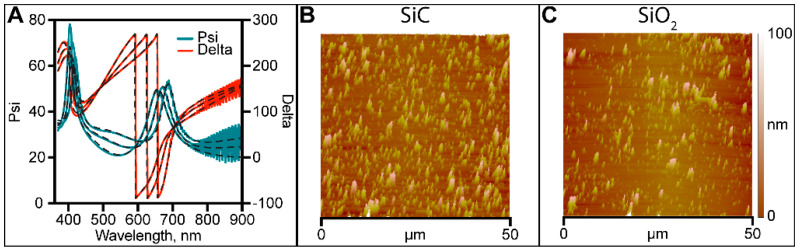
(**A**) Ellipsometric measurement of a-SiC blank sample. (**B**) a-SiC blank sample 50 μm × 50 μm 3D topographic map. (**C**) SiO_2_ blank sample 50 μm × 50 μm 3D topographic map.

**Figure 3 membranes-11-00535-f003:**
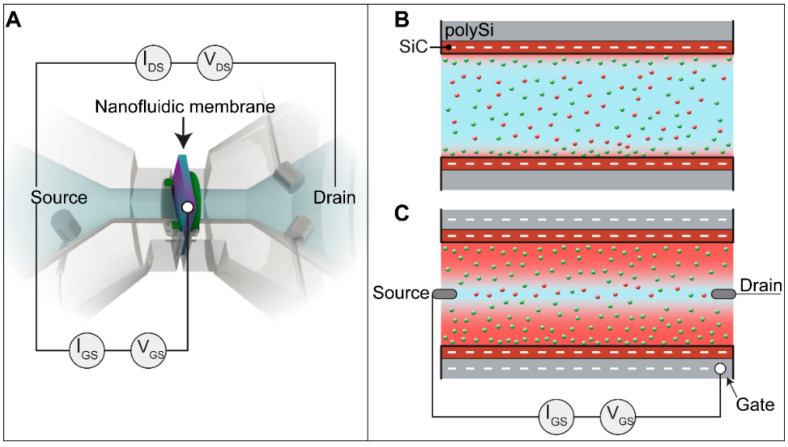
(**A**). To perform the I-V and conductance measurements, transmembrane potential V_DS_ is applied between the source electrode and drain electrode in the reservoir, and gate potential V_GS_ is applied between the source electrode in the reservoir and the buried gate electrode of the membrane. (**B**) Telescopic illustration of the nanochannel wall and the buried gate electrode. (**C**). Application of gate potential V_GS_ that polarizes the surface of the nanochannel wall, leading to alteration of the conductance of the charged species and, consequently, modulation of the ionic current I_DS_.

**Figure 4 membranes-11-00535-f004:**
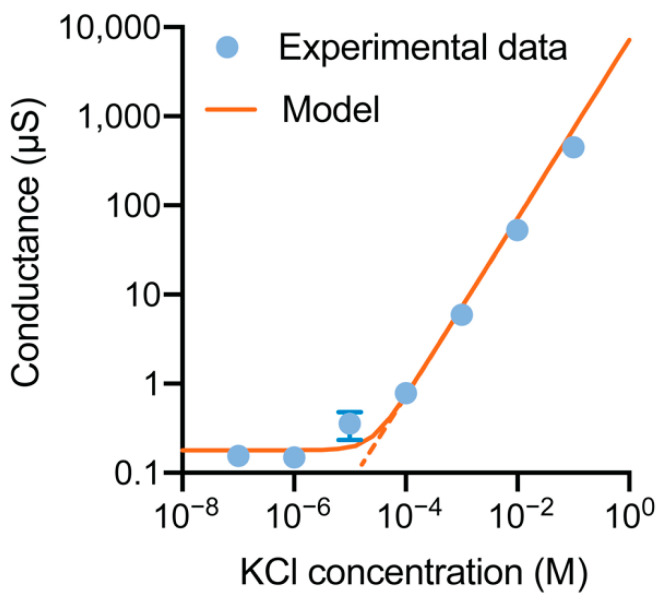
Measured (points) and calculated (curve) ionic conductance across the nanochannels versus the electrolyte concentration. The dashed line represents the bulk prediction, which deviates from the experimental data in the low ionic concentration region. The standard deviation of the measurements was calculated based on three membranes, tested three times each.

**Figure 5 membranes-11-00535-f005:**
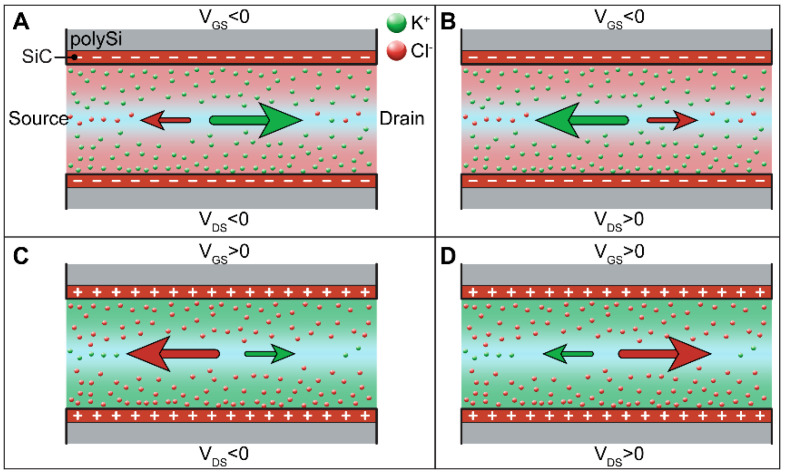
Ionic current I_DS_ when gate potential V_GS_ and transmembrane potential V_DS_ are applied. Red points represent Cl^−^ ions, while green points represent K^+^ ions. (**A**) V_DS_ < 0 V and V_GS_ < 0; (**B**) V_DS_ > 0 V and V_GS_ < 0; (**C**) V_DS_ < 0 V and V_GS_ > 0; (**D**) V_DS_ > 0 V and V_GS_ > 0.V_DS_ < 0 V attracts positive potassium ions K^+^ to the drain electrode, whereas V_DS_ > 0 V attracts negative chloride ions Cl^−^ to the source electrode. V_GS_ > 0 V draws Cl^−^ to the nanochannel wall, creating a negatively charged EDL. Consequently, the number of negative ions in the nanochannel increases, and positive ions K^+^ are repelled. V_GS_ < 0 V attracts K^+^ to the nanochannel walls, leading to an increased number of K^+^ and a decreased number of Cl^−^.

**Figure 6 membranes-11-00535-f006:**
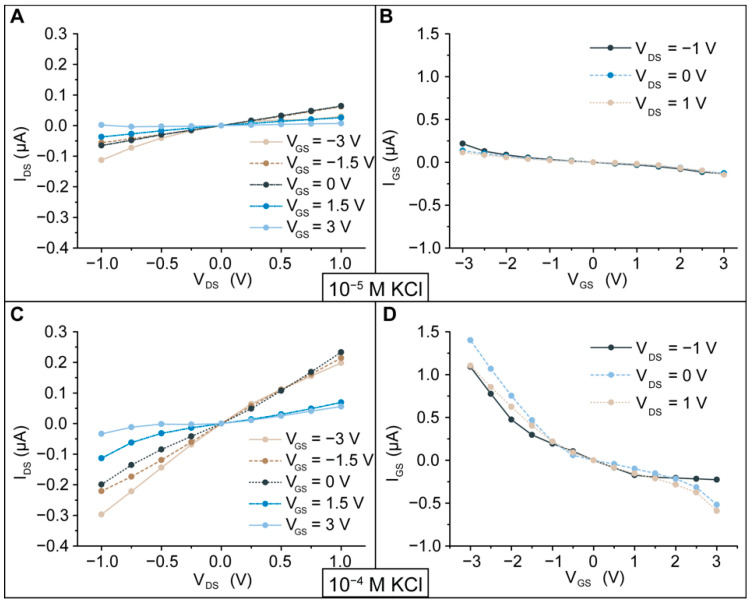
I-V (I_DS_ vs. V_DS_) ionic transfer characteristics and leakage current (I_GS_ vs. V_GS_) characteristics of the nanofluidic membrane measured for 10^−5^ M KCl (λ~96 nm) and 10^−4^ M KCl (λ~30 nm). (**A**) I_DS_ vs. V_DS_ measured for 10^−5^ M KCl. (**B**) I_GS_ vs. V_GS_ measured for 10^−5^ M KCl. (**C**) I_DS_ vs. V_DS_ measured for 10^−4^ M KCl. (**D**) I_GS_ vs. V_GS_ measured for 10^−4^ M KCl.

**Figure 7 membranes-11-00535-f007:**
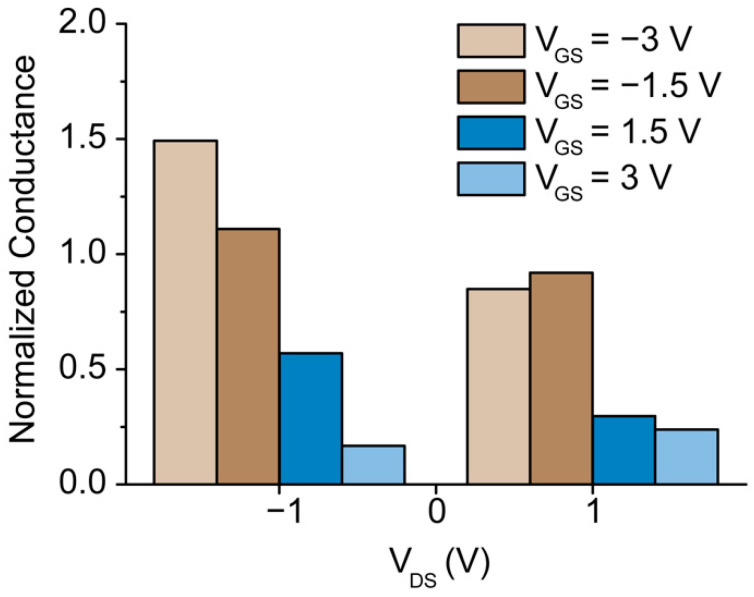
Normalized conductance for 10^−4^ M KCl, defined as the ratio of the conductance after application of the gate voltage to the conductance measured with a floating gate for V_DS_ = −1 V and V_DS_ = 1 V.

**Figure 8 membranes-11-00535-f008:**
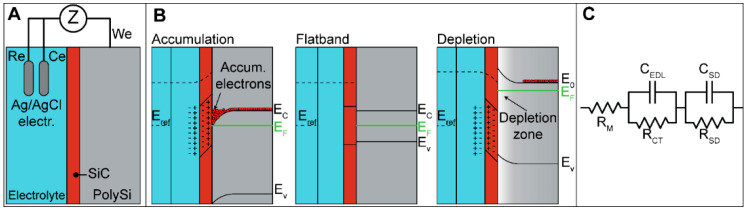
Blank setup measurements. (**A**) Blank sample immersed in the KCl electrolyte. (**B**) Band diagram of a layered structure composed of the semiconductor poly-Si, a dielectric, and an electrolyte. Accumulation for V_GS_ < 0, flat band for V_GS_ = 0 V, and depletion for V_GS_ > 0 V. (**C**) Equivalent circuit used to fit the impedance data. R_M_ is the electrolyte resistance, R_SD_ and C_SD_ are the resistance and the capacitance associated with the dielectric coating, R_CT_ is the charge transfer resistance, and C_EDL_ is the Helmholtz capacitance due to the EDL.

**Figure 9 membranes-11-00535-f009:**
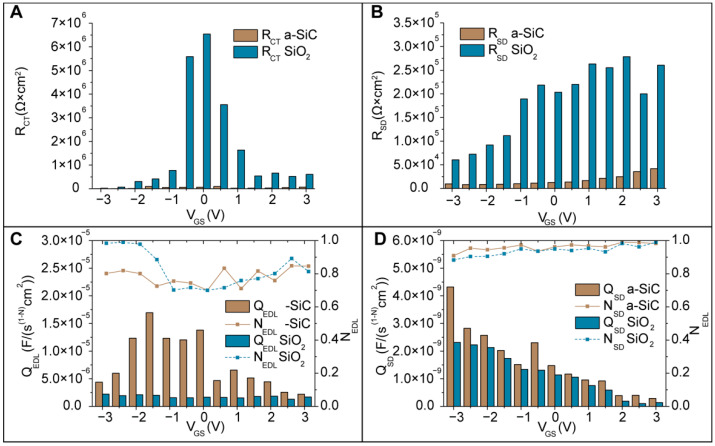
Comparison between the fitted values for a-SiC and SiO_2_. (**A**) R_CT_ charge transfer resistance; (**B**) R_SD_ resistance of the polysilicon; (**C**) Q_EDL_ constant phase element and its n_EDL_ factor associated with the EDL; (**D**) Q_SD_ constant phase element and its n_SD_ factor associated with the polysilicon.

**Figure 10 membranes-11-00535-f010:**
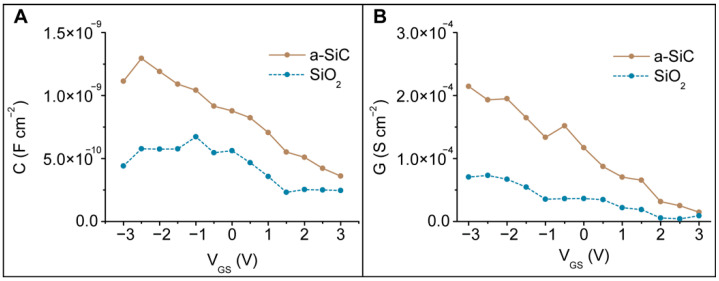
(**A**) Capacitance-voltage (C-V) and (**B**) conductance-voltage (G-V) data at a high frequency of 10^5^ Hz evaluated on blank devices.

**Figure 11 membranes-11-00535-f011:**
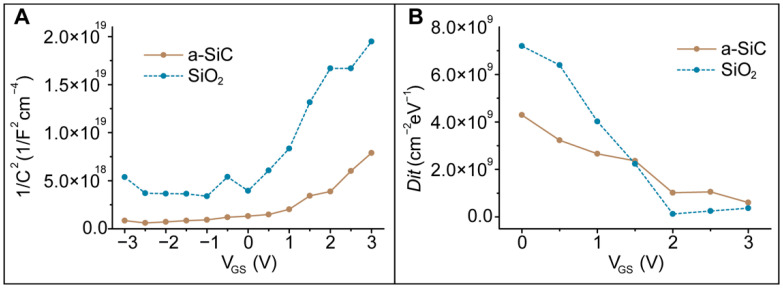
(**A**) Mott–Schottky plot for a-SiC and SiO_2_. Flat band potential evaluation for a-SiC and SiO_2_ electrodes in 10−4 M KCl. (**B**) Extracted density of interface states for a-SiC and SiO_2_.

## Data Availability

The data presented in this study are available within the manuscript.

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
