# Peer review of "Silicon Carbide-Gated Nanofluidic Membrane for Active Control of Electrokinetic Ionic Transport"

_membranes, 2021, doi:10.3390/membranes11070535_

Round 1

Reviewer 1 Report

The authors of the manuscript, titled "Silicon carbide gated nanofluidic membrane for active control of electrokinetic ionic transport," study a novel membrane for electrokinetic control of ionic species. The electrokinetic properties of the membrane are in depth characterized, analyzed and modeled based on measurements and using band theory. I really enjoyed reading the manuscript and the work appears very solid. Overall, the manuscript is well written, clearly structured and the results are presented in clear diagrams. When the minor issues mentioned below will be resolved, I can recommend the manuscript for publication.

Minor Issues:

  1. line 316: Equation (1) symbol d is not described.
  2. line 324: the pH should be specified along with the surface charge density
  3. Figure 5 caption: Red points represent Cl- ions, while green points represent K+ ions => must be consistent with the colors used in the figure.
  4. Figure 5C/D: It is unclear why, in the case of VGS > 0, the arrows for the currents of K+ and CL- ions are the same size.
  5. Figure 6: the Debye length for the concentration 0.1 mM KCL should be specified.
  6. Page 14: It remains unclear how RCT, RSD, QEDL, and QSD have been determined. An SI would help.
  7. Line 616: 10^5 Hz

Reviewer 2 Report

In this work, a biocompatible electrode-embedded nanofluidic channel membrane designed for electrofluidic applications is proposed. The authors investigated the ability of a nanofluidic membrane to electro-statically control charged ions in a monovalent aqueous KCl solution through the application of a potential to the embedded electrode. This approach implies a uniformly distributed doped polysilicon electrode buried in membrane nanochannels coated with biocompatible a-SiC, which may lead to a new type of silicon implantable nanofluidic systems for tunable drug delivery and personalized medicine. The topic is interesting and the solid results were got in experiments. There are some points to be addressed in the revision of the manuscript.

  1. The authors have presented that the materials used to fabricate nanofluidic devices are crucial to the biocompatibility of the devices and cited some papers such as ref. 15, 42-45, and 47. Nevertheless, the reason why a-SiC is a suitable material for long term implantable applications is obscure. The state-of-art of a-SiC nanochannels should be described in details to clearly figure out the novelty of this work.
  2. Line 86, silicon nitride (S3N4) should be silicon nitride (Si3N4).
  3. The author should give the schematic diagram to show the fabrication flow of the nanofluidic membrane since the present description is a little confusing.
  4. The AFM topographic map for SiO2 should also be presented in Fig. 2.
  5. In the figure caption of Fig. 5, “Blue points represent Cl- ions, while red points represent K+ ions.” There are no blue points in the figure. The author should check the figure and revise the figure caption.
  6. EIS has been performed in section 3, why the frequency ranged from 10-2 Hz to 105 Hz is used for measurements? And besides, the Nyquist plots should be presented before Fig. 9 to clearly figure out RM.
  7. To compare dielectric performance, the author conducted the same measurement on an insulator, SiO2. Could the author explain why use SiO2 in this work? How about the performance for other semiconductors?
